# Qualitative understandings of the persistent use of traditional contraceptive methods using the socio-ecological model among older reproductive-age women in Khulna, Bangladesh

Mashiat Abedin, Mohammad Bellal Hossain◉*

Department of Population Sciences, University of Dhaka, Dhaka, Bangladesh

* bellal@du.ac.bd

## Abstract

### Background

The continued use of Traditional Contraceptive Methods (TCM) among Bangladeshi women of reproductive age (aged 35–49 years) poses a significant public health issue. Existing research in Bangladesh suggests that women in this age group use TCM more than their younger counterparts. However, the reason for the higher TCM use among Bangladeshi women of reproductive age is yet to be explored—the current study aimed to understand the use of TCM among Bangladeshi women aged 35 years.

### Methods

This qualitative study employed purposive sampling from the Khulna district to conduct ten in-depth interviews with women aged 35–49 years and seven key informant interviews with family planning service providers. A socio-ecological model was adopted for this study. Data were collected in January 2024. The interviews were audio-recorded and transcribed verbatim. Thematic data analysis was performed.

### Results

At the individual level, women's poor knowledge and fear of the side effects of modern contraceptive methods, perceptions related to the effectiveness, risks, and benefits, ease of use, and cost shaped the use of TCM. Interpersonal factors included the influence of spouses, mothers-in-law, and peer groups on the women. Community norms and beliefs are also pivotal in this regard. Institutional factors include providers' attitudes, health facility-related issues such as distance from the house, waiting in queues, unavailability of products, and policy-level influences, such as the lack of

**Data availability statement:** The data underlying the results presented in the study are reposited in the Open Science Framework and available from https://osf.io/k326b/. (DOI 10.17605/OSF.IO/K326B).

**Funding:** The author(s) received no specific funding for this work.

**Competing interests:** The authors have declared that no competing interests exist.

an updated policy that promotes TCM use among older women of reproductive age in Bangladesh.

## Conclusion

The complex interplay of various factors shapes the use of TCM in Bangladesh. Therefore, comprehensive reproductive health education programs should be considered to enable women to make informed choices about contraceptive use and switch from traditional to modern contraceptive methods, such as long-acting methods. This will ultimately lead to improved reproductive health outcomes.

## Introduction

The continued use of traditional contraceptive methods (TCM) among older reproductive-aged women (ages 35–49) in Bangladesh poses a significant public health issue. Despite achievements in promoting the use of modern contraceptives, many older women remain dependent on traditional practices such as withdrawal, periodic abstinence, and folk methods [1]. In Bangladesh, the contraceptive prevalence rate (CPR) among married women is 64%, with 54.7% and 9.3% using modern methods and TCM, respectively. Over the years, the use of TCM has been higher among women aged 35–49 compared to their younger counterparts. It was the highest among those aged 40–44 in 2022 [1].

Women aged 35–49, representing 15.73% of the female population in Bangladesh, face a higher risk of unintended pregnancies [2,3]. One-third of pregnancies in Bangladesh are unplanned, of which 63% are experienced by women aged ≥ 35 years [3]. TCM users have a higher proportion of unintended pregnancies than modern method users [3]. Unsafe abortions are often associated with unintentional conception [3,4]. Bangladesh is experiencing an epidemiological transformation, with older women facing elevated levels of obesity and non-communicable diseases, such as hypertension and diabetes [1]. These conditions are likely to increase maternal health complications and adverse birth outcomes [5–7].

Existing research in Bangladesh has focused on overall contraceptive practices among women, overlooking variations in contraceptive behavior and preferences across age groups [8–17]. While these studies identified determinants of TCM use among Bangladeshi women, such as age, spousal interaction, and socioeconomic and spatial factors, they did not fully capture the broader spectrum associated with TCM use. Factors such as media and visits by family planning workers have been studied [10,13]; however, other critical demand-side issues, including women's perceptions, and supply-side factors, such as healthcare facility-related issues and policies, remain to be explored for the use of TCM in Bangladesh. The findings of studies conducted worldwide imply that various issues influence the decision to use TCM. From the demand perspective, women's views on modern contraceptive methods are frequently influenced by a lack of knowledge, misconceptions, and a preference for TCM, which is believed to be less complicated, for example, with no side effects and

more convenient to use compared to modern ones. On the supply side, factors such as poor standards of family planning services, lack of counseling on side effects, and challenges with logistics (such as lengthy commutes to clinics or contraceptive shortages) contribute to the use of TCM [18,19]. However, these factors have not yet been studied in relation to the use of TCM in Bangladesh. This research gap is particularly evident in quantitative studies, which often fail to investigate the reasons why older women use TCM. Qualitative exploration of the complex justifications, beliefs, and lived realities that support TCM is imperative. To date, no qualitative studies in Bangladesh have addressed these concerns. Therefore, targeted investigations are needed to consider the intricate dynamics of demand and supply issues regarding why TCM use is higher among older women of reproductive age in Bangladesh.

This gap hinders the formulation of effective policies and initiatives that address the needs of older women of reproductive age and undermines efforts to achieve the Sustainable Development Goals (SDGs), particularly those related to health and gender equality [20]. This study explores why the use of TCM is higher among older reproductive-aged women in Bangladesh, thereby bridging the knowledge gap and guiding effective policies and practices.

## Materials and methods

### Theoretical orientation of the study

This study adopts the Socio-Ecological Model (SEM), which provides a comprehensive framework for examining the interplay between individuals and their surrounding social environments. It includes several tiers of influence—individual, interpersonal, community, institutional, and policy—on behavior [21]. Various factors, including knowledge and perception, interpersonal relationships, community norms and beliefs, access to and quality of care, and national policies, influence contraceptive behavior [18,22,23]. This model provides a comprehensive understanding of the factors affecting TCM use. Although multilevel factors have been studied regarding contraceptive use in general and modern methods, specifically in the context of Bangladesh [16,17] using quantitative methods, SEM has not been applied to address the dynamics of TCM use among older women in Bangladesh using a qualitative approach. Therefore, this study employed SEM to understand the use of TCM among women of reproductive age in Bangladesh.

### Research strategy and design

This study adopted a qualitative research strategy based on interpretive epistemology [24] to understand the use of TCM by exploring the unique perceptions of Bangladeshi women and the meanings they ascribed to their experiences. In addition, the study adopted a cross-sectional research design because the data were collected at a single point in time [24].

### Study area

The study was conducted in Khulna District, which was purposively selected because of its high use of TCM. The use of TCM has consistently been high in this division. For example, the rates over the last decade have varied from 10.6% in 2011 and 10.7% in 2014 to 12.5% in 2017 and 10.2% in 2022, respectively [1]. This site allowed us to capture a range of experiences and practices in a context where TCM use is prevalent. While we acknowledge that Bangladesh exhibits regional variations in healthcare access, education, and cultural norms (e.g., Sylhet shows distinct patterns in contraceptive use, and urban Dhaka has different health-seeking behaviors), the aim of this qualitative study was not to achieve national representativeness but to generate in-depth insights from the context of high TCM reliance. We purposefully selected married women aged 35–49 who reside in three sub-districts or Upazilas of Khulna district: Rupsha, Batiaghata, and Dumuria Upazila.

### Study participants

The study population consisted of older reproductive-aged women, that is, 35–49 years of age, who were using TCM, such as periodic abstinence or withdrawal, and family planning service providers and program managers working at

various levels under the Directorate General of Family Planning, Ministry of Health and Family Welfare, Government of the People's Republic of Bangladesh. Older women aged 35–49 years were included, as the use of TCM is consistently higher in this age group than in younger women, despite the nationwide promotion of modern methods over time [1]. This study purposively focused on older women of reproductive age (35–49 years) to better understand the persistent reliance on traditional contraceptive methods (TCM) among those who had completed or were near completing their fertility. This age group was selected because prior research has shown that persistence and switching contraceptive use behaviors differ across the life course, and older women may be more entrenched in long-term TCM practices [15]. Consequently, these women tend to constitute a persistent group of TCM users whose beliefs and decisions have not been previously explored. We aimed to fill this gap in the literature by examining the multifaceted aspects (as defined by the Socio-Ecological Model) that influence the choice of TCM. Data were collected from 10 older reproductive-aged women using purposive sampling. Most women were aged 35–39 years with more than one living child. All women were users of the periodic abstinence or safe period method, except for one who practiced both the safe period and withdrawal methods. Most were Muslims, and the majority had an education level of higher secondary or below. Seven participants were family planning service providers and program managers working at various levels of the Directorate General of Family Planning. This included assistant directors, Upazila family planning officers, medical officers, family welfare visitors, and family welfare assistants. Family planning service providers and program managers were recruited from the Maternal and Child Welfare Centre (MCWC), the Upazila Family Planning Office, and the Union Health & Family Welfare Centre (UH&FWC) in Bangladesh, which provide family planning services.

## Data collection

Data collection for this study commenced on 27/01/2024 and ended on 05/02/2024. Qualitative data were collected from older reproductive-aged women through face-to-face in-depth interviews (IDIs) and from family planning service providers and program managers through key informant interviews (KIIs) using a semi-structured interview guide. These three types of respondents were interviewed because SEM-related contraceptive use requires data from both demand and supply sides. We employed IDIs with women and KIIs with service providers, rather than focus group discussions (FGDs), because of the sensitive nature of the topic. FGDs may have constrained participants from openly sharing their personal and potentially stigmatized experiences in a group setting. In contrast, IDIs provided a confidential space to elicit deeper narratives, while KIIs ensured the inclusion of service provider perspectives. Two separate semi-structured interview guides were developed based on extensive literature reviews and refined in response to the emerging interview results. Therefore, semi-structured interview guides were used to structure the interviews while providing flexibility. The interviews lasted 30–45 minutes. The interviews were conducted in Bengali, which is the native language. Ten IDIs and seven KIIs were conducted. The sample size was determined using the data saturation concept, which is a typical criterion for qualitative studies. This sample size is consistent with qualitative research guidelines, where saturation is often reached within 12–20 interviews [25,26]. We found a recurrence of major themes, indicating that the level of saturation had been reached, as evidenced by the absence of any new codes. In our case, after the 8th interview with women and the 6th KII, no new substantive themes appeared, indicating that thematic saturation had been achieved. Furthermore, a qualitative study prioritizes in-depth and rich data over the size of the sample, and similar prior studies on reproductive health have employed samples of a comparable size in Bangladesh and elsewhere [27,28].

## Data analysis

The study adhered to the thematic analysis approach developed by Braun and Clarke [29] to provide an organized framework for finding, examining, and summarizing the themes in the qualitative data. The audio recordings of the interviews were transcribed and read numerous times to ensure a deeper understanding of the participants' responses. A hybrid coding approach was adopted to integrate both the inductive and deductive methods. While the inductive codes resulted from the participants' narratives, the deductive codes were guided by the components of SEM and previous research. An initial

coding framework was generated after examining the subsections of the transcripts, and new codes were assigned if they could not be classified using the initial framework. The coding was performed manually. The codes and themes were arranged in Microsoft Excel spreadsheets, enabling transparent and explicit comparisons and facilitating theme development. The procedure involved both authors, who collaborated to address any variations in comprehending and interpreting the codes, thereby strengthening the credibility of the study analysis. The authors had no notable differences in the identified codes. The codes were then arranged according to broad themes that encapsulated the participants' perspectives and experiences. The themes were revised and refined to ensure that they adequately depicted the data. Meaningful participant quotes were used to validate these themes, providing evidence of the participants' lived experiences. The Standards for Reporting Qualitative Research (SRQR) checklist was used for methodological rigor where applicable [30].

## Ethical considerations

This study was approved by the Department of Population Sciences of the University of Dhaka (serial no. CAR 2023/002). The study participants, all adults, were provided with information about the research's purpose, the questions to be asked, and their freedom to refuse or discontinue the interview at any time during the process. Before each interview, participants were asked if they agreed to participate and informed that the session would be recorded, while maintaining confidentiality and anonymity. The participants provided verbal consent, which was audio-recorded with the approval of the ethics board. All transcripts and excerpts in the findings section contained anonymous identifiers to protect the participants' identities.

## Results: Factors shaping traditional method use

Following the SEM, this study identified individual, interpersonal, community, institutional, and policy-level factors influencing the use of TCM among older reproductive-aged women in Bangladesh. The Table 1 summarizes the dominant themes that emerged as factors shaping the use of traditional contraceptive methods.

**Table 1. Themes and sub-themes that shaped the use of traditional contraceptive methods.**

| Individual Level | 1. Lack of knowledge about modern methods<br>2. Fear of side-effects of modern methods<br>3. Knowledge of traditional methods<br>4. Perceived effectiveness of traditional methods<br>5. Perceived ease of using traditional methods<br>6. Perceived low risk of traditional methods<br>7. Perceived cost of modern methods<br>8. Perceived health benefits<br>9. Perceived embarrassment<br>10. Exposure to mass media |
|---|---|
| Interpersonal Level | 1. Spousal influence<br> • Opposition to modern methods<br> • Cooperation to use traditional methods<br> • Spousal attitude<br>2. Influence of mother-in-law<br>3. Peer influence |
| Community Level | 1. Community approval of contraceptive use<br>2. Common fear of medical complications<br>3. Religious beliefs |
| Institutional Level | 1. Providers' attitude<br>2. Irregular visits<br>3. Quality of service<br>4. Healthcare center-related issues |
| Policy Level | 1. Lack of updated policy |

**Individual level factors**

**Lack of knowledge regarding modern methods.** Many participants were unfamiliar with modern methods and lacked understanding of what they were and how to apply them. A participant with higher secondary education revealed her knowledge of modern methods, leading to widespread fear of potential side effects.

*"I don't understand that much. I haven't used [modern methods]. Many people talk about Copper-T, but I do not understand them; I do not have a good knowledge about this."* (IDI-9)

Another participant with a primary education stated a similar level of understanding of modern contraceptives. This demonstrates that, despite education levels, women lacked knowledge about modern methods, partly due to their low awareness and partly due to the widespread myth and fear of potential side effects of modern methods, which further prevented them from becoming familiar with these methods.

*"Let alone using them, I don't even know about them [modern methods]. I never thought about using them or learning about them…"* (IDI-10)

Providers also reported poor awareness among women regarding modern methods, leading to misconceptions about these methods and the choice of TCM. This emphasizes the pressing need for targeted awareness initiatives to overcome misunderstandings, increase knowledge, and enable women to make informed choices regarding reproductive health.

*"A lot of them have misconceptions [about modern methods]. They do not know about it [modern methods]. That's why they do this [safe period]. They don't know what the other options are."* (KII-6)

**Fear of side-effects of modern methods.** Fear of adverse effects was a major reason why women in Bangladesh chose TCM. Women who had observed or experienced harmful consequences, such as dizziness, weight gain, and nausea, linked these issues directly to modern methods, prompting them to use TCM, which they believed to be safer. For example, a woman with a higher education level stated that even after knowing modern methods like pills, injectables, and ligation, she preferred not to use them because she was afraid of the risks and side effects.

*"I see my sister-in-law having many problems. She uses injections. She has a vision problem; she does not see properly. She has irregular periods every three to four months. She now sees almost nothing. She has become fatter. She had stopped receiving the injections. Pills also deteriorate the body regularly, such as dizziness, vomiting, and other health problems…"* (IDI-4)

The providers reported that fear of the adverse side effects of modern methods acted as a barrier to the utilization of modern methods, such as oral pills and injectables. They attempted to alleviate these fears and anxieties by emphasizing that adverse effects tend to be brief as the body adapts to hormonal changes. However, their efforts frequently failed to influence women's beliefs, resulting in the continued use of TCM as a safer choice.

*"For example, if you use it for the first time, it will take some time to adapt to your body. This will take two to three months. There were some changes in the menstrual period. These are hormonal methods; therefore, if they are absorbed in the body, the ability to have children is hindered. This may be the reason why they think something is wrong with their bodies. If they have a little patience, they can understand that their ability will return. That's why it is called the reversible method."* (KII-1)

**Greater understanding of traditional contraceptive methods.** Women have a greater understanding of TCM, such as the safe-days method or periodic abstinence, confidently identifying the "fertile" and "infertile" days of their cycle.

*"I am counting safe days and keeping track of the date of my menstruation. If my period starts on the 10th of the month, I keep track of when it starts… so that's how I maintain it. A week before the start of menstruation is safe." (IDI-2)*

However, the providers felt that the women were not fully aware of the safe days and miscalculated their fertile days. They further highlighted that even educated women were not fully aware of how to use TCM, making them vulnerable to unwanted pregnancies.

*"Let me share an experience. A project manager at a reputed organization and his wife followed the [safe period]. The days he thinks are safe are dangerous. This is how they conceived a baby. But we know that it is a dangerous time." (KII-1)*

**Perceived effectiveness of traditional methods.** Users of the safe-days method were confident in their ability to calculate safe days because their regular monthly cycles were accurately calculated. The absence of unplanned pregnancies strengthened their satisfaction with the method's effectiveness, demonstrating a sense of control and confidence in managing their reproductive health. Consequently, they saw no reason to transition to modern methods.

*"Yes, it works well; we do it. I believe I can maintain this well. It's straightforward for those who understand, and it's difficult for those who do not understand how to do it [counting safe period]." (IDI-2)*

However, opposing viewpoints from healthcare experts emphasized this divergence, stating that many women had difficulty maintaining correct counts, resulting in a substantial failure rate and unwanted pregnancies. This disparity highlights the discrepancy between individuals' self-assessed capabilities and professionals' concerns about the reliability of TCM, underscoring the need for enhanced education and counseling to bridge this gap.

*"They can't do it [count safe days] properly. You must maintain it. However, since they cannot keep account [of the safe days], the failure rate is quite high. As a result, they have many children." (KII-6)*

**Perceived ease of use of traditional methods.** The participants felt that using TCM was more convenient than modern methods, indicating a preference based on their perceived ease of use and minimal disruption to their daily tasks. For example, some women did not remember taking pills regularly, while others found it challenging to use condoms or missed due dates for injectables. This perception of TCM as an easy alternative highlights a significant barrier to the adoption of modern methods, implying that strengthening accessibility and constant follow-up for modern contraceptives may be critical in promoting their use among women of reproductive age.

*"There is no unnecessary trouble (with TCM), like forgetting to take pills, forgetting to take injections, I can't remember the time, and I can't always keep an account. There is no hassle. If you consume pills or use any other method, you must stand in line at the hospital and waste time. Also, you must wait for the workers [to visit home]; sometimes they do not have it and do not visit regularly." (IDI-1)*

**Perceived low risk of TCM.** The participants had a low-risk perception of TCM use. They believed that their method was less harmful and had no unintended consequences, such as unplanned pregnancies. They felt safe adopting these methods because they had no firsthand experience with undesirable outcomes or had heard of negative outcomes.

*"It seems this is good for my health; there is no risk [of unintended pregnancy]. I believe this is good for me. I'm fine with Allah's mercy. I trust it [safe period], and if I keep an account [of safe period], there is no problem." (IDI-1)*

Furthermore, health practitioners noted a significant gap in women's awareness, stating that many were ignorant of the larger health hazards linked to TCM, including sexually transmitted infections (STIs). This assumption regarding TCM's safety, along with the fear of modern methods, led to the decision to use TCM, indicating a need for comprehensive counseling to correct misconceptions and encourage safer contraceptive behaviors.

*" I think those using traditional methods are unaware of the health risks. If they knew about HIV-AIDS or other sexually transmitted diseases, they would not have used the Azal method [withdrawal] or other [TCM]. In this case, we generally recommend the use of condoms to avoid health risks." (KII-2)*

**Perceived cost of modern methods.**  Women from low-income households viewed modern methods as financially inaccessible. They believed that buying contraceptives, such as pills, from local pharmacies or stores, when not available in government health facilities, would be an extra expense for their households. Furthermore, they believed that permanent methods, such as ligation, necessitated expensive surgeries, suggesting their limited understanding of the availability of cost-subsidized, long-acting, and permanent options. This notion was supported by the participants' inclination toward cost-free TCM, such as recording safe periods, which they believed to be more affordable and appropriate for their socioeconomic situation than modern methods.

*"Also, we're not so rich, and we do not have much money to do that [ligation], so I think this method [safe period] is good for people like us." (IDI-1)*

**Perceived health benefits of the intervention.**  Many women believe that TCM offers fewer health hazards than modern contraception, which they associate with a variety of adverse effects, such as weight gain and other physical discomforts. This perception was supported by individual and family experiences in which older generations used TCM without harmful health impacts, strengthening the view that TCM was safe.

*"If I use something else, it will not adjust to my body, she said. Therefore, I use it [safe period]. Our grandmothers, aunts, and mothers used it [safe period] and did not have any problems with it; they are healthy. I have not gained weight. I have remained the same before and after marriage." (IDI-4)*

*"I talk to people, but I never hear them say they have a disease or anything due to this [using safe period]." (IDI-1)*

Furthermore, providers typically refrained from suggesting modern methods to women with pre-existing health issues, which supported their choice of TCM.

*"They [women] think that it [modern method] can increase their health risks at this age. When they cross 30/40, they may have high blood pressure and diabetes. Therefore, we do not suggest modern methods for patients with diabetes or high blood pressure. Therefore, they are more prone to traditional methods. And they try to avoid modern methods." (KII-2)*

**Perceived embarrassment.**  The embarrassment associated with modern contraception, such as condoms and injections, underlines longstanding societal beliefs that discourage open family planning discussions. The participants mentioned how societal criticism led to their unwillingness to use modern methods. For example, disposing of condoms was considered a shameful deed that might result in condemnation from family members, particularly from in-laws. Similarly, visiting medical facilities for injectables was regarded as publicly reporting their contraceptive use, exposing them to potential scorn and disgrace. The stigma associated with contraceptive use fostered the adoption of TCM, which was viewed as a more discreet alternative.

*"Before my marriage, I used to see that people would make fun of those who used to go to healthcare centers for family planning methods. I think it is about condoms. You must throw it away or dispose of it after use, and I have my family to consider. What if my sisters-in-law and mother-in-law see it while throwing it away? It's such a shame."* (IDI-2)

**Exposure to mass media.** Exposure to mass media plays a critical role in increasing awareness and shaping women's attitudes toward different contraceptive methods, prompting them to consider modern contraception. The participants admitted to having seen or heard family planning messages on television or radio before; however, they had not encountered many recently. This explains women's lack of knowledge of modern methods, especially long-acting and permanent ones, leading to the continued use of TCM.

*"Yes. I have seen it [family planning messages] on TV, for example, having one child is enough, whether it is a girl or a boy. I remember they used to show dramas on TV as well, but not anymore."* (IDI-9)

## Interpersonal level

**Spousal opposition to modern methods of childbearing.** Partner opposition to modern methods was a barrier to their adoption, typically driven by concerns about adverse consequences, including obesity and other health issues. The spouse's refusal to let his wife choose between pills or injections symbolized a larger sociocultural effect, in which male opinions profoundly shaped reproductive decisions.

*"He [husband] does not like pills or injections. He says that these [pills and injections] will make me obese and there will be many diseases."* (IDI-4)

In response, the providers acknowledged the importance of religious figures such as Imams who wield communal power. By utilizing Imams to promote the benefits of modern methods, they hoped to leverage their authority to resolve spousal opposition and increase approval.

*"The men of the family can be explained by the Imam (who leads the prayers in a mosque). The men will not listen to me, but they accept the words of the Imams. We give them a list of names and ask them to motivate and discuss such things."* (KII-4)

**Cooperation is required to use traditional methods.** Most participants reported that their husbands cooperated and supported the use of TCM, especially when they observed that modern methods, such as oral pills, had adverse effects.

*"You know, my husband saw that there is a problem with that [pills], and it [safe period] is safe. Therefore, he accepted it. We discussed [safe period] after my sister told me about it [safe period], and he told me to follow it."* (IDI-8)

**Spousal attitude.** Spouses' positive attitudes toward TCM facilitated its use among women in Bangladesh. The husbands felt that TCM was more effective, so they supported their wives and encouraged them to use it. This scenario illustrates how partner involvement and mutual satisfaction with TCM led to its continued use, demonstrating the importance of spousal support in reproductive choices. It highlights how partner involvement and mutual satisfaction with TCM contributed to its continued use, underscoring the significance of spousal support in reproductive decision-making.

*"My husband says that this [safe period] is good. I feel physically fine. Additionally, we have spent many years in this manner, and he is content with it. He believes there is nothing wrong with it. He also said that I had a problem with pills, but that this one is safer. We are better." (IDI-1)*

**Influence of the mother-in-law.** Family members, especially mothers-in-law, play a crucial role in shaping women's choices and decisions regarding family planning methods. Opposition to modern methods by mothers-in-law often leads to the use of TCM.

*"Honestly speaking, she doesn't like this [family planning]. She says don't we have [children] in our times... What's wrong with having [children] if Allah gives you? If Allah gives a mouth, He will feed them as well… We have not used such useless things. We have raised five or six children; you can do it too. After marriage, my mother-in-law told me not to use these things." (IDI-1)*

However, providers remarked that mothers-in-law's dominance waned over time, and many no longer opposed the use of modern contraception.

*"The role of the family is important… The mothers-in-law have become quite aware of things, even in our village. We had eight to ten children, and we had no problems. Nowadays, this example doesn't work" (KII-2)*

**Peer influence.** Peer groups, such as sisters and friends, shape the use of TCM among women in Bangladesh. Some participants mentioned their sisters as sources of information and support for the use of TCM. However, even when their peers disapproved, the participants continued to use TCM because of their low-risk perception.

*"My elder sister uses this method as well. She advised me to use this method. She also told me that if I follow the rules properly, there will be no problem." (IDI-8)*

## Community level

**Community approval of contraceptive use.** The community's approval of contraceptive use has grown dramatically, creating a more accepting setting for women to select contraceptive methods that satisfy their requirements, whether modern or traditional.

*"I don't think nowadays anyone has any headache or anything about anyone. People use methods that are beneficial for them or suit their individual needs. No one has discussed this. They use whatever is convenient for them… I see most people are consuming pills. That's what I hear the most." (IDI-1)*

Even family planning workers, who once faced prejudice and were often disliked by communities, are now welcomed and regarded as trusted, demonstrating a positive shift in community norms. However, lingering negative opinions about modern methods exist at the individual level.

*"Society plays a crucial role, and it has changed over time. There was a time when people could not even say they worked in family planning because of various reactions. The FWAs could not go home, and people chased them out. Now, people welcome them with open arms. However, a limited part is negative; it was there before and will remain so." (KII-1)*

**Common fear of medical complications.** Fear of medical complications and false beliefs, such as implants breaking inside the body during daily activities or IUDs migrating inside the body, have led to the discontinuation of modern methods and a switch to TCM use. These fears and misconceptions were frequently perpetuated and reinforced through community interactions among neighbors and relatives, affecting individual choices to avoid using modern contraceptives.

*"If you undergo an operation such as ligation, you must undergo many cuts. What if the cut is not well stitched? Many people find it difficult to come home and work after the operation… Sometimes, it may lead to cancer as well… Also, there can be problems in the uterus and cervix, and sometimes they must be cut off as well. So, I am scared. My neighbor told me not to do it." (IDI-1)*

Health professionals acknowledged the existence of preconceptions about modern methods among women, thus recognizing the need to create tailored educational programs to overcome these misunderstandings.

*"Many people think IUDs mix with blood and heart, go inside, and they fear that it will be displaced." (KII-6)*

**Religious beliefs.** Religion plays an important role in the use of TCM and the non-use of modern methods among older women of reproductive age in rural areas. Only one participant was Hindu, while the rest were Muslim. According to Muslim women, modern methods such as injectables, implants, and ligation were heinous and prohibited because they involved alterations to the body that were regarded as inconsistent with religious teachings. Therefore, they rejected modern methods in favor of traditional ones to ensure religious observance.

*"According to my religion, it is not right to take any pills or anything. Using a condom or a pill is a sin. They [religious leaders] also think the same because it is a sin to use a pill..." (IDI-8)*

Women, on the other hand, justified their use of TCM in comparison to the sinful aspects of modern religious methods. They believed that TCM, such as the safe period method, was natural, acceptable, and non-sinful, and aligned better with their beliefs than Western methods. Thus, it was legitimate to use it according to religion.

*"Why will it [safe days] be forbidden? I don't do any sin. Is this my sin? No! We are doing many things in the world; we are doing sinful things. I don't think it is sinful. How do those who use injections and have irregular periods and sporadic bleeding pray?" (IDI-9)*

Furthermore, providers highlighted the religious reasons for women's opposition to the use of modern methods, such as divine punishment and retribution in the afterlife. As a result, they attempt to cope by enlisting spiritual figures, such as imams, to clarify myths and advocate modern contraception.

*"Allah will not forgive them; that part of the body will be burnt in Hell. In this case, we approach the heads of mosques, such as the Imam, to motivate the people and share benefits [of using modern methods]." [KII-6]*

### Institutional level

**Providers' attitude.** The participants expressed mixed feelings about the behaviors of their healthcare providers. Some women found them accommodating, but others mentioned feeling angry if they ignored the providers' advice. The inconsistency in professional attitudes led to a loss of faith and poor experiences for women, particularly when healthcare

workers encouraged them to adopt specific methods, such as implants or ligation, without providing an adequate reason or acknowledging their fears.

*"Sometimes they behave well, and at other times they get angry… they insist on keeping [products]… When I said no, she got angry. She furiously told me, "What are you doing? You don't want to take the pill? You don't want to perform ligation? What is this? You have two children." They behave so strangely, do not explain properly, do not come to us regularly, but still get angry!" (IDI-1)*

In response, program managers believed that they delivered thorough training to improve their communication skills and foster good rapport and trust, especially at the grassroots level. They emphasized the use of Behavior Change Communication (BCC) to develop the skills necessary to address individual issues, convey accurate information, and provide respectful and patient-friendly care.

*"We have BCC [Behavioral Change Communication] strategies. We train them to communicate effectively with others. However, more training is needed because they repeatedly perform the same tasks. That's why sometimes we give them refresher training, so their work is better." (KII-3)*

**Irregular FWAs visits.** Some participants reported receiving inadequate and irregular home visits from Family Welfare Assistants (FWAs). FWAs are grassroots healthcare workers who provide reproductive and maternal health services to communities in their designated locations, including contraceptive distribution, counseling, and, if necessary, referrals to higher-level healthcare professionals. They serve as a critical link between women and the larger health system, and irregular attendance can result in missed opportunities for education, contraception, and prompt referrals. Consequently, the women felt less supported, which contributed to their continued use of TCM rather than modern options.

*"Earlier, they used to come and call everyone out, "Khala/Chachi (aunts)." Now they don't come that often." (IDI-9)*

**Quality of service.** Some women expressed skepticism and apprehension because of previous poor experiences or perceived maltreatment by health professionals, preventing them from seeking family planning services at health centers. Fear of encountering practitioners' harsh attitudes and satisfaction with TCM prompted women to avoid healthcare centers entirely. This finding demonstrates that strengthening interpersonal care components is critical for women's use of modern family planning options.

*"No, I didn't go [to healthcare centers]. If those who come to our homes behave like this, then the ones [at the healthcare centers] may behave like this, and there may even be people in higher positions. I am afraid of how they will behave toward me. And I am satisfied with my method [safe period], so I do not go." (IDI-1)*

**Healthcare center-related issues.** Women reported issues at the healthcare center, such as the distance from their homes, waiting in lengthy queues, and product unavailability, which led to frustration and contributed to the non-use of modern methods of menstrual hygiene management. The hassle and scarcity of resources deter women from accessing essential services.

*"If you go there, you will find that there are often long lines and many problems with the serials…. They often claim that no more products are available. They give a date and say, "Come later on this day, we don't have it right now, you can take it next time." It's so troublesome." (IDI-8)*

On the other hand, providers emphasized patient-friendly care by creating an enabling, courteous, and individualized atmosphere that focused on the well-being of those receiving healthcare, making them feel at ease, valued, and adequately cared for.

*"Our union-based healthcare centers work as a team. Those who go to the field are always in contact with those at the centers. They also considered whether any complications were associated with the use of any method. If any complications arise after using a family planning method, the government will cover the associated costs. This is how we have been able to create a patient-friendly environment." (KII-1)*

The key informants further explained how they ensured that women who sought family planning services at health centers had full access to them. They outlined the financial considerations for long-acting and permanent methods of contraception services provided to women, as per government instructions.

*"If someone comes to get clinical services in that FWC (Family Welfare Center) … we pay a transportation cost. Once 60 Taka is given to them, it is used for follow-up. Three follow-ups were performed monthly. The amount for the three months is also provided. They are informed when the FWV (Family Welfare Visitor) motivates them to undergo the procedure. And it is mentioned in the citizen charter." (KII-1)*

### Policy level

**Lack of updated policy.** Family planning initiatives in Bangladesh have emphasized the promotion of modern contraceptive methods to combat the growing population and promote the well-being of mothers and children. Traditional methods have been a part of the general family planning scenario but are rarely promoted in mainstream policy because of high failure rates, often resulting in unintended pregnancies.

*"In fact, according to government policy, we talk about the modern method, not the traditional method. And the reason is because you don't know what will happen if they use it, such as unintended pregnancy" (KII-5)*

Additionally, the providers discussed ongoing initiatives to revise national policy, including consultations with stakeholders to accommodate varied family planning needs, doorstep care, and greater knowledge of women's needs. Given the number of older reproductive-aged women using TCM in Bangladesh, instead of sidelining traditional methods in policy discourse, it is imperative to adopt an inclusive approach and raise awareness of the negative consequences of using TCM among women.

*"The national policy that was in place in 2012 is now being further modified. Meetings have also been held with various stakeholders and are expected to be completed this year. Family planning activities have been incorporated into national policy through the coordination of all stakeholders. School education, door-to-door services, the complexity of service provision, and the needs of women- everything has been included." (KII-1)*

## Discussion

This study showed that individual, interpersonal, community, institutional, and policy factors shape the use of traditional contraceptive methods among Bangladeshi women aged 35–49. At the individual level, women's knowledge of TCM and modern methods, as well as their perceptions of effectiveness, risks and benefits, side effects, ease of use, and cost, influenced their use of TCM.

The study found that participants were familiar with the benefits of family planning but lacked awareness of modern methods. Instead, they monitored their "safe" days, exhibiting a low-risk perception. This finding is similar to that of a study in Indonesia, where educated women knowledgeable about family planning and fertile periods were more likely to use TCM [31]. Similarly, previous research in Bangladesh showed that women's knowledge of monthly cycles enhanced the possibility of practicing contraception, as they could plan and act during their fertile periods to avoid conception [17]. Furthermore, women believe that the TCM they use is effective because they have been using it for a long time without encountering any unintended consequences, which is consistent with findings from other studies worldwide [19,32]. This perspective contrasts with providers' concerns about the frequent failures of TCM, particularly due to a lack of sufficient understanding of the reproductive cycles. The fears of the providers accord with empirical evidence from Bangladesh, where unintended pregnancies were greater among TCM users due to their failure rates [3]. This emphasizes the importance of comprehensive counseling in bridging the perception gap and encouraging informed decisions by women.

Most participants found TCM easier to use than modern methods, as they did not need to remember to take pills or visit health facilities to receive injections. This result is congruent with the findings of other studies [19,32–34]. Moreover, our participants found using modern methods shameful, which is concurrent with some studies where participants reported that they were 'embarrassed' to buy a condom [34–37]. The participants had very low-risk perceptions related to the use of TCM, considering its reliability and safety in preventing unwanted pregnancy. A possible reason for this perception is that these women did not experience unintended pregnancies. Some accepted the risk but believed it could be minimized correctly, while others felt that modern methods were riskier. This is in line with a study in Turkey [19] and differs from other studies in which TCM users expressed concerns about the risk of unintended pregnancy [34].

Unwanted side effects of modern methods, such as becoming overweight, infertility, dizziness, nausea, vomiting, fatigue, diminished appetite, and irregular periods experienced by women and their family members or neighbors, lead to their discontinuation and use of TCM [31,38,39]. Various studies have confirmed this phenomenon globally [15,40]. Like the current study, prior research in Bangladesh showed that irrespective of primary, secondary, or higher educational level of women, use of modern methods was lower [12,16] due to the fear of potential side-effects of pills, injectables, implants, etc. [12]. This suggests that such fear, without receiving balanced advice on the advantages of the methods and their potential side effects, can deter educated women from using modern contraceptive methods. This calls for more scientific research to develop modern methods with minimal side effects, ensuring that women's health remains unaffected. Traditional methods were cost-free, while modern methods were primarily purchased from the private sector (57%), with 37% supplied by the public sector in Bangladesh. For example, 14% of users obtain pills from government field workers, whereas 59% obtain pills from pharmacies/drugstores [1]. When women do not have access to pills due to irregular visits by workers or stockouts, they resort to pharmacies, resulting in costly prescriptions. Women may consider the private sector to be more courteous, accessible, and trustworthy. A coordinated approach is necessary to strengthen public family planning services, including health infrastructure and professional training. A similar outcome has been reported previously [41–43]. Likewise, withdrawal users in Turkey reported that this method was not costly, whereas modern methods were [19,32].

Exposure to mass media has influenced the adoption of TCM among older women. Women reported knowledge about family planning through television advertisements and dramas, but they observed a decrease in such coverage. Previous studies have found that TV and radio are essential for delivering information on family planning in Bangladesh, although exposure to family planning through mass media remains constrained [44,45]. Additionally, evidence indicates that women in Bangladesh who come across family planning content on television tend to use contraception more than those who do not [14]. The most efficient way to convey information about contraception is through mass media, which is consistent with other studies [46]. Mass media messages weighing the benefits and risks of using TCM versus modern methods can help women make informed choices. Policymakers should consider alternative media, including social networking sites, to address and resolve women's fears regarding the adverse effects of modern methods, while highlighting the risks of TCM, such as unplanned pregnancies.

In Bangladesh, patriarchal gender roles frequently influence women's decisions regarding reproduction, particularly among older women of reproductive age, who are expected to follow the wishes of their family or husbands. At the interpersonal level, male partners' opposition to modern methods hinders women from using them [19,35]. Supportive husbands favored the use of TCM due to its perceived benefits. This mutual decision-making among spouses was in line with other studies [19,32]. Bangladeshi literature consistently shows that spousal dynamics play an important role in shaping contraceptive decisions [17]. The current study confirms that men continue to be the dominant decision-makers regarding contraceptive use, implying that women's ability to choose modern contraceptives is limited. This highlights the long-standing need to increase male participation in family planning programs. Mothers-in-law's aversion to modern methods impedes the adoption of oral pills, injectables, and so on, as they have substantial power in a typical household in Bangladesh, including reproductive decisions. This scenario is consistent with the findings of previous studies [47–49]. However, prior research has implied that living with mothers-in-law may promote the use of modern methods and TCM in India, as well as modern methods such as pills, injectables, and implants in Nepal and Bangladesh [50]. This can be attributed to context-specific factors, such as disparities in educational levels, access to family planning information, and the implementation of program methods in different locations. For instance, mothers-in-law might be more practical and supportive of modern methods if they have been involved in family planning programs and activities. However, they might also put pressure on women to uphold such ideals in circumstances where conventional values are deeply ingrained and childbearing obligations are evident. This highlights the importance of context-specific family planning interventions that involve husbands and mothers-in-law in discussions, such as courtyard meetings. In this study, peer support played a key role, as women were more inclined to opt for the method utilized by their social circle, as they valued their opinions and experiences. This finding is consistent with prior research from Bangladesh, which has shown how social groups affect the current use of contraceptives [51]. In Bangladesh, women's peer networks are primarily dependent on family members, like mothers, sisters, and kin from in-laws. If a woman experiences adverse reactions to a contraceptive, she shares the situation with her peers, sometimes in exaggerated versions, making other women afraid to use similar types of contraceptives. However, if the peer group supports family planning and promotes modern methods, women approve of specific contraceptives. This highlights the potential application of social networking strategies to disseminate modern contraceptive information through interpersonal interactions [51,52].

The use of contraceptives is common; however, community and religious beliefs act as barriers to the uptake of modern methods, facilitating TCM use. Religion affects contraceptive use among Bangladeshi women, with modern methods often seen as forbidden, while TCM is accepted [8,9,11,12,16,18,53]. Women believed that using modern methods, especially permanent ones, was wrong in terms of religion, dreading the aftermaths such as being denied a proper religious burial or facing eternal punishment [54]. Furthermore, cultural myths and beliefs negatively affected the uptake of modern contraceptives, aligning with published research worldwide [18,22,35,55–58]. This underscores the significance of involving influential individuals, such as Imams, in addressing the acceptability of long-acting permanent methods (e.g., IUD, ligation) within the context of religion and dispelling misconceptions.

Contraceptive use was shaped by institutional-level factors, such as a lack of proper healthcare infrastructure, absence of quality care and services, and poor behavior of family planning service providers, which calls for client-centered care services. This is consistent with the results of a study conducted in West Africa [59]. According to a Bangladeshi study, when modern methods of contraception were not available in local health centers, women could be forced to buy them privately or commute a long distance to a different clinic, both of which might be expensive and difficult to access. This was especially problematic for impoverished women who might not have the autonomy or resources to travel alone to access abortion services. Therefore, the use of modern methods, such as long-acting contraceptives, has decreased [4]. Other studies have reported that users prefer services to be available at convenient times, with shorter waiting periods, and located near their residences [37,60]. Providers' undesirable attitudes, such as yelling and neglecting concerns about side effects, further hamper service utilization [61]. Furthermore, commodity availability issues underscore the urgency

of improving supply chain monitoring and achieving greater consistency in consumption estimations in relation to product availability [61,62].

Bangladesh has achieved success in family planning despite challenges such as illiteracy, lower gender equality, and poverty. The 4th Health, Population, and Nutrition Sector Program (HPNSP) 2017–22 targeted raising the CPR from 62% to 75% by 2022, focusing on underperforming regions such as Sylhet and Chattogram. Bangladesh Family Planning 2030 (FP2030) pledges to expand the supply, access, and adoption of modern contraceptives [1,63]. Existing policies, such as the Bangladesh Population Policy 2012, aimed to increase CPR by 72% by 2015 [64]. However, they must be updated to incorporate the complexity of family planning and shifting demographic patterns. These policies emphasize modern methods, which are much more reliable, safer, and more effective than TCM, which has adverse consequences such as unintended pregnancies and sexually transmitted diseases. This underscores that policies inadequately address the requirements of 35–49-year-old women and thus the need for a transition beyond a "*one-size-fits-all*" approach to family planning that includes tailored strategies to meet the distinct requirements for the reproductive well-being of Bangladeshi women aged 35–49.

While the current research organizes factors shaping TCM use across the five tiers of SEM, our results also show intricate interactions among education, income, and gender dynamics that transcend these boundaries. In contrast to the widely held belief, educated women with higher household incomes were more likely to use TCM. Their choice of method was frequently influenced by concerns about adverse effects, prior undesirable experiences with modern methods, or familial circumstances within the home. Furthermore, women with limited resources often depended on TCM because of a lack of financial accessibility, availability, awareness, and misconceptions. A similar pattern was observed in a previous study conducted in Bangladesh [15]. Therefore, there is a need to develop TCM user-inclusive policies that incorporate tailored counseling for older reproductive age women, including those who experience adverse outcomes of modern methods, targeted mass media initiatives to highlight the risks of using TCM, dispel misconceptions related to modern methods, and community-based programs to combat familial opposition and local norms and beliefs. Additionally, providers require retraining to address women's concerns about modern methods with sensitivity and care.

### Strengths and limitations

This study specifically explored the traditional contraceptive behaviors of older women of reproductive age in Bangladesh using SEM. It provides a comprehensive overview of the process by which women choose TCM. This has significant implications for family planning initiatives targeting the intended population. This study had several limitations. First, we acknowledge that the findings from Khulna may not accurately reflect the experiences of women in other regions of Bangladesh, such as rural Sylhet or urban Dhaka, which differ in terms of education, healthcare access, and cultural practices. A multi-district approach could have enhanced the transferability of the findings. However, our purposive choice of Khulna was methodologically appropriate for exploring TCM practices in a district where its use is particularly widespread. As with most qualitative studies, our goal was to provide depth and contextual understanding rather than statistical generalizability of the results. Nevertheless, the insights generated can inform future research across diverse regions and provide a foundation for comparative inquiries. Second, while the relatively small sample size limits the generalizability of the study, the purpose of this qualitative study was not to generate population-level estimates but to provide an in-depth, context-specific understanding. In line with qualitative research principles [65], our findings are analytically generalizable and transferable to similar contexts, offering valuable conceptual insights for future policy and research. Third, we acknowledge that the study relied on IDIs and KIIs without FGDs, which may have provided additional insights into collective cultural dynamics. Nevertheless, the use of interviews was methodologically appropriate, given the sensitivity of the subject matter. Fourth, while our focus on older women allowed us to examine the persistence of TCM use among those nearing the end of their reproductive years, future research should include younger women to capture intergenerational and life-course dynamics. Such studies would complement our findings by examining how social, institutional, and interpersonal barriers differentially

shape reliance on TCM across age groups. Finally, the opinions of the participants' husbands were not explored. Nonetheless, it was sufficient to provide a genuine picture of TCM use among older reproductive-aged women.

## Conclusion

This study employed a socio-ecological model to explore traditional contraceptive methods among women aged 35–49 years old. Individual, interpersonal, community, institutional, and policy factors, such as poor knowledge of modern contraceptive methods, perceived ease of use and risk related to the use of TCM, fear of side effects of using modern methods, familial influence, community norms and beliefs, providers' attitudes, quality of services, and other health-related issues such as distance from the house, waiting in queues, unavailability of products, and lack of mass media coverage shaped TCM use. Consequently, family planning services should be strengthened to promote modern methods, address related fears, enhance knowledge, and highlight the risks of TCM through comprehensive counseling targeted at women of older reproductive age. Community based initiatives like courtyard meetings should involve influential individuals such as husbands, mothers-in-law, and religious figures, as their participation is critical in disseminating and normalizing modern methods in society. Family planning strategies can incorporate realistic media content for older women of reproductive age, presenting examples of women who have successfully switched to modern methods while raising awareness of the potential risks of using TCM. Policies must ensure that service providers receive up-to-date and evidence-based training to deliver empathetic, client-centered care that acknowledges women's worries, debunks myths and misconceptions regarding adverse effects, and enhances trust in modern contraceptives, especially IUDs, implants, and sterilization. Reproductive health programs should help women make informed choices and switch from TCM to modern methods, such as long-acting ones. This will enhance women's reproductive health and promote gender equity, thereby contributing to sustainable development in the country.

## Acknowledgments

The authors thank all participants for their consent to participate in this study.

## Author contributions

**Conceptualization:** Mashiat Abedin, Mohammad Bellal Hossain.

**Data curation:** Mashiat Abedin.

**Formal analysis:** Mashiat Abedin, Mohammad Bellal Hossain.

**Investigation:** Mashiat Abedin, Mohammad Bellal Hossain.

**Methodology:** Mashiat Abedin, Mohammad Bellal Hossain.

**Supervision:** Mohammad Bellal Hossain.

**Validation:** Mohammad Bellal Hossain.

**Writing – original draft:** Mashiat Abedin.

**Writing – review & editing:** Mashiat Abedin, Mohammad Bellal Hossain.

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
