## [Decision Letter · Decision Letter 0]

15 May 2025

Dear Dr. Hossain,

Thank you for submitting your manuscript to PLOS ONE. After careful consideration, we feel that it has merit, but we invite you to submit a revised version of the manuscript that addresses the points raised during the review process.

We look forward to receiving your revised manuscript.

Kind regards,

Zahid Ali Memon, MSc, MPH, PhD

Academic Editor

PLOS ONE

Journal Requirements:

2. In the online submission form, you indicated that [Data cannot be shared publicly because of ethical issues. However, a formal request for data can be made to the corresponding authors of this manuscript.].

Reviewers' comments:

Reviewer's Responses to Questions

**Comments to the Author**

1. Is the manuscript technically sound, and do the data support the conclusions?

Reviewer #1: Yes

Reviewer #2: Yes

2. Has the statistical analysis been performed appropriately and rigorously?

Reviewer #1: No

Reviewer #2: N/A

3. Have the authors made all data underlying the findings in their manuscript fully available?

Reviewer #1: Yes

Reviewer #2: Yes

4. Is the manuscript presented in an intelligible fashion and written in standard English?

Reviewer #1: No

Reviewer #2: Yes

Reviewer #1: The study gives useful information about the use of Traditional and Complementary Methods (TCM) in Bangladesh. However, it could be made stronger by including a wider range of participants, explaining the methods in more detail, and offering clearer policy suggestions. Using the Socio-Ecological Model (SEM) works well, but the study would benefit from also looking at how different factors like gender, income, and education levels interact.

Reviewer #2: This is a well-designed and clearly written study that offers original insights into the persistent use of traditional contraceptive methods among older reproductive-age women in Bangladesh. While existing literature has broadly discussed contraceptive practices in the region, this study contributes a valuable angle by applying a socio-ecological model to understand the specific choices and constraints of women aged 35–49.

The research question is clearly articulated and appropriately addressed. The qualitative design is well-suited to the topic, and the methods—purposive sampling, in-depth interviews, and key informant interviews—are adequately described and ethically sound. Participants are sufficiently contextualised, and the conditions under which data were collected are transparent.

The findings are credible and logically structured, and the study rightly emphasises the layered influences—from individual to policy level—that shape contraceptive use. However, the conclusion would benefit from clearer articulation of the implications for policy and practice. While the message is largely sound, refining how it is conveyed would increase the paper’s clarity and potential impact.

The use of the SRQR checklist is appropriate and strengthens the paper’s rigour. References are relevant and current, with no obvious omissions. The abstract accurately reflects the content of the paper.

Overall, this manuscript is a valuable contribution for both researchers and policymakers working on reproductive health in South Asia. It is worthy of publication.

**Do you want your identity to be public for this peer review?** For information about this choice, including consent withdrawal, please see our Privacy Policy

Reviewer #1: **Yes: ** Hina Najmi

Reviewer #2: **Yes: ** Sadiq Bhanbhro

---

## [Author Response · Author response to Decision Letter 1]

9 Sep 2025

Editor’s Comments

Methods

1. Sample Size and Justification

The study conducted 10 in-depth interviews (IDIs) with women and seven key informant interviews (KIIs) with service providers. The study relies solely on in-depth interviews (IDIs) and key informant interviews (KIIs), missing the opportunity to include focus group discussions (FGDs). This limits the depth of social and cultural insights into TCM. On the other hand, the small sample size raises concerns about saturation (whether no new themes emerged) and generalizability.

● The authors should justify why this sample size was deemed sufficient, possibly referring to qualitative research guidelines (e.g., Braun & Clarke’s thematic analysis) or prior similar studies.

Response:

We thank the reviewer for this insightful comment. We have revised the Methods and Discussion sections to clarify our methodological choices more effectively. Specifically, we have explained that FGDs were not used due to the sensitive nature of the topic, where participants might have felt uncomfortable sharing personal experiences in a group setting. Instead, IDIs provided a confidential environment to capture rich narratives, complemented by KIIs with service providers (Please see lines 148-152).

We have also justified our sample size with reference to established qualitative research guidelines (Guest et al., 2006; Godfrey et al., 2011; Braun & Clarke, 2021; Hossain & Hossain, 2023), noting that saturation is typically achieved within 12–20 interviews. In our study, no new themes emerged after the 8th IDI and the 6th KII, indicating thematic saturation (Please see lines 157-165). Finally, we clarified in the Discussion that the purpose of this qualitative study is not statistical generalizability, but rather analytical generalizability and transferability, in line with the traditions of qualitative research (Lincoln & Guba, 1985) (Please see lines 727-732).

2. District Selection and Representativeness

● The study was conducted in Khulna District, selected due to its high TCM use. However, Bangladesh has significant regional variations in healthcare access, education, and cultural norms (e.g., Sylhet has lower contraceptive use). Moreover, the findings may not reflect the experiences of women in other regions (e.g., rural Sylhet or urban Dhaka). A multi-district approach could enhance transferability.

Response:

We thank the reviewer for this important observation. We have revised both the Methods (please see lines 104-114) and Discussion (please see lines 718-727) sections to clarify our rationale for selecting the study area more effectively and to acknowledge the limitations. Specifically, we note that Khulna District was purposively chosen because of its high use of TCM, which provided a relevant context for exploring the study objectives. The title of the study has been refined to highlight the unique geographical focus, while also addressing the issue of regional representation. This modification enhances clarity and aligns with qualitative research standards that prioritize context-rich insights over generalizability (please see lines 1-3). At the same time, we acknowledge that regional variations in healthcare access, education, and cultural norms mean that the findings may not be fully transferable to other settings, such as Sylhet or Dhaka. We have highlighted this as a limitation in the Discussion (please see lines 718-727) and suggested that future research adopt a multi-district approach to enhance transferability (please see lines 721-722).

3. Study Participants

The study focuses on older reproductive-age women (35–49 years) to understand persistent traditional contraceptive method (TCM) use, but excludes younger women (15–34 years), despite evidence showing:

● Higher unintended pregnancy rates in younger women (Noor et al., 2011; Khan et al., 2022).

● Lower modern contraceptive prevalence (MCPR) among adolescents and young women in Bangladesh (BDHS 2022).

● Greater reliance on TCM due to stigma, lack of access, or misinformation (Kamal & Islam, 2010; Kundu et al., 2022).

● In addition to this, the use of SEM examines multi-level influences, yet the study ignores a key demographic facing unique interpersonal (parental control) and institutional (youth-unfriendly services) barriers.

Response:

This study purposefully targeted older reproductive-age women (35-49 years) because recent evidence indicated that this demographic relied substantially more on Traditional Contraceptive Methods (TCM) compared to younger women, despite the nationwide promotion of modern methods over time. These women tend to constitute a persistent group of TCM users whose beliefs and decisions have not previously been explored. We aimed to fill this gap in the literature by examining the multifaceted aspects (as defined by the Socio-Ecological Model) that influence their choice of TCM (Please see lines 129-131). Our findings showed that this demographic relied on TCM due to a lack of knowledge & fear of side effects of modern methods, as well as spousal influence, and barriers such as community beliefs and institutional factors, including a poor attitude among providers. Therefore, attention should be given to the persistent use of TCM among this specific age group of women. We agree that younger women face distinct interpersonal and institutional challenges; therefore, we recommend that future investigations broaden the age bracket to include adolescents and young women (please see lines 735-739).

Analysis

1. Thematic approach and use of software

The authors followed Braun & Clarke’s framework, but more detail is needed on:

a. Coding process: Was it inductive/deductive? How were discrepancies resolved?

b. Inter-coder reliability: Were multiple researchers involved in coding to reduce bias?

c. Software use: Mention if tools like NVivo were used for transparency.

Response:

We appreciate the opportunity to elaborate on and expand our thematic analysis. We have updated the Methods section to incorporate the coding technique used in this study, as well as the collective resolution of coding inconsistencies by the authors. The coding was done manually. Codes and themes were arranged in structured spreadsheets, enabling transparent and explicit comparisons and facilitating theme development. As a result, meticulous documentation and collaboration among researchers established a rigorous methodology (please see lines 168-186).

2. Thematic Presentation of Results

The results are clearly structured using the Socio-Ecological Model (SEM), addressing factors at the individual, interpersonal, community, institutional, and policy levels. The inclusion of rich participant quotes strengthens the thematic findings, for instance, highlighting concerns such as fear of side effects and spousal opposition. However, some contradictions emerge in the narratives: while many women expressed satisfaction with Traditional and Complementary Methods (TCM), providers emphasized the high failure rates associated with these methods. It would be helpful to present participants’ sociodemographic characteristics and include data on intended versus unintended pregnancies, as the current results provide limited insight into prior experiences with unplanned pregnancies. Additionally, the analysis could be strengthened by exploring how variables such as education level, income, and urban or rural residence intersect with TCM use, as the current discussion remains relatively broad.

Response:

Thank you for your thoughtful comment. We have modified the discussion section to address the contradiction between women's satisfaction with TCM and the providers' emphasis on its high failure rates (Please see lines 574-582). We have already described the participants’ sociodemographic characteristics in the Study Participants section of the Methods section (please see lines 116-141). However, we have not collected detailed data on unplanned pregnancies but asked participants whether they perceived unplanned pregnancies as a risk of TCM. As mentioned in the Results section, they had a very low risk perception of TCM, partly because they had not experienced any such outcomes (please see lines 294-309, 590-93). Additionally, we have discussed how variables such as education level, income, and others influence the use of TCM throughout the manuscript.

3. Discussion

1. Cross-Referencing and Contextualization

The discussion effectively links findings to prior studies (e.g., Turkey, Indonesia) but could:

o Compare more directly with Bangladeshi literature (e.g., Kamal & Islam, 2010; Hossain et al., 2018).

o Address contrasting evidence (e.g., some studies show mothers-in-law promote modern methods).

o Policy gaps and recommendations need to be highlighted

Response: Thank you for your insightful comments. We have modified the discussion section to include related Bangladeshi studies, discussed contradictory results in the available literature, and provided more precise policy recommendations that reflect our findings.

Overall Review of the research:

The study provides valuable insights into the utilization of Traditional and Complementary Methods (TCM) in Bangladesh. However, it could be strengthened by including a wider range of participants, providing more detailed explanations of the methods, and offering more precise policy recommendations. Using the Socio-Ecological Model (SEM) is effective, but the study would benefit from also examining how various factors, such as gender, income, and educational levels, interact.

Response: Thank you for your overall review. We have modified our text as required to accommodate your valuable suggestions.

Reviewer’s Comments

Reviewer #1:

Comment: The study provides valuable insights into the utilization of Traditional and Contraceptive Methods (TCM) in Bangladesh. However, it could be strengthened by including a wider range of participants, providing more detailed explanations of the methods, and offering more precise policy recommendations. Using the Socio-Ecological Model (SEM) is effective, but the study would benefit from also examining how various factors, such as gender, income, and educational levels, interact.

Response: We thank the reviewer for providing such helpful and critical input. We appreciate your acknowledgement of the application of the Socio-Ecological Model (SEM) to explore the use of TCM among older reproductive-age women in Bangladesh. In response to your suggestion:

● Wider range of participants: This study purposefully targeted older reproductive-age women (35-49 years) because recent evidence indicated that this demographic relied substantially more on Traditional Contraceptive Methods (TCM) compared to younger women, despite the nationwide promotion of modern methods over time. These women tend to constitute a persistent group of TCM users whose beliefs and decisions have not previously been explored. We aimed to fill this gap in the literature by examining the multifaceted aspects (as defined by the Socio-Ecological Model) that influence their choice of TCM (please see lines 129-131). Our findings showed that this demographic relied on TCM due to a lack of knowledge & fear of side effects of modern methods, as well as spousal influence, and barriers such as community beliefs and institutional factors, including a poor attitude among providers. Therefore, attention should be given to the persistent use of TCM among this specific age group of women. We agree that younger women face distinct interpersonal and institutional challenges; therefore, we recommend that future investigations broaden the age bracket to include adolescents and young women (please see lines 735-739).

● Detailed explanation of methods: We have revised the Methods section to provide greater clarity regarding our rationale for using in-depth interviews and key informant interviews, sample size, and how data were coded and analyzed in the study.

● Clearer policy suggestions: We have revised our text to include more straightforward policy suggestions that reflect our findings (Please see lines 749-764).

● Interaction of gender, income, and education with TCM use: We appreciate this critical suggestion. We have now reflected on how factors such as gender dynamics, income, and education intersect with TCM use in the Discussion section, drawing on women’s narratives and prior literature.

Reviewer #2:

Comment: This is a well-designed and clearly written study that offers original insights into the persistent use of traditional contraceptive methods among older reproductive-age women in Bangladesh. While existing literature has broadly discussed contraceptive practices in the region, this study contributes a valuable angle by applying a socio-ecological model to understand the specific choices and constraints of women aged 35–49.

The research question is clearly articulated and appropriately addressed. The qualitative design is well-suited to the topic, and the methods—namely, purposive sampling, in-depth interviews, and key informant interviews—are adequately described and appear to be ethically sound. Participants are sufficiently contextualized, and the conditions under which data were collected are transparent.

The findings are credible and logically structured, and the study rightly emphasizes the layered influences—from individual to policy level—that shape contraceptive use. However, the conclusion would benefit from more precise articulation of the implications for policy and practice. While the message is sound, refining how it is conveyed would increase the paper’s clarity and potential impact.

The use of the SRQR checklist is appropriate and strengthens the paper’s rigor. References are relevant and current, with no obvious omissions. The abstract accurately reflects the content of the paper. Overall, this manuscript is a valuable contribution for both researchers and policymakers working on reproductive health in South Asia. It is worthy of publication.

Response: We appreciate your informative and encouraging comments. In response to your suggestion for the conclusion, we rewrote and updated the final section of the paper to clearly articulate the policy implications of our findings (Please see lines 749-764).

---

## [Editor Report · Decision Letter 1]

13 Oct 2025

Qualitative understandings of the persistent use of traditional contraceptive methods using the socio-ecological model among older reproductive-age women in Khulna, Bangladesh

PONE-D-25-12194R1

Dear Dr. Hossain

We’re pleased to inform you that your manuscript has been judged scientifically suitable for publication and will be formally accepted for publication once it meets all outstanding technical requirements.

An invoice will be generated when your article is formally accepted. Please note, if your institution has a publishing partnership with PLOS and your article meets the relevant criteria, all or part of your publication costs will be covered. Please make sure your user information is up to date by logging into Editorial Manager at Editorial Manager®  and clicking the ‘Update My Information' link at the top of the page. For questions related to billing, please contact billing support .

Kind regards,

Zahid Ali Memon, MSc, MPH, PhD

Academic Editor

PLOS ONE
---

## [Editor Report · Acceptance letter]

PONE-D-25-12194R1

PLOS ONE

Dear Dr. Hossain,

I'm pleased to inform you that your manuscript has been deemed suitable for publication in PLOS ONE. Congratulations! Your manuscript is now being handed over to our production team.

Kind regards,

on behalf of

Dr. Zahid Ali Memon

Academic Editor

PLOS ONE